# Precision mitochondrial medicine

## Patrick F. Chinnery[1,2] 

[1]Department of Clinical Neurosciences, School of Clinical Medicine, University of Cambridge, Cambridge Biomedical Campus, Cambridge, UK and [2]Medical Research Council Mitochondrial Biology Unit, University of Cambridge, Cambridge Biomedical Campus, Cambridge, UK

genetic polymorphism; genetic risk score; genetics; genomics; metabolic diseases

**Author for correspondence:**
Patrick F. Chinnery,
Email: pfc25@cam.ac.uk

## Abstract

Mitochondria play a key role in cell homeostasis as a major source of intracellular energy (adenosine triphosphate), and as metabolic hubs regulating many canonical cell processes. Mitochondrial dysfunction has been widely documented in many common diseases, and genetic studies point towards a causal role in the pathogenesis of specific late-onset disorder. Together this makes targeting mitochondrial genes an attractive strategy for precision medicine. However, the genetics of mitochondrial biogenesis is complex, with over 1,100 candidate genes found in two different genomes: the nuclear DNA and mitochondrial DNA (mtDNA). Here, we review the current evidence associating mitochondrial genetic variants with distinct clinical phenotypes, with some having clear therapeutic implications. The strongest evidence has emerged through the investigation of rare inherited mitochondrial disorders, but genome-wide association studies also implicate mtDNA variants in the risk of developing common diseases, opening to door for the incorporation of mitochondrial genetic variant analysis in population disease risk stratification.

## Impact statement

Mitochondria are cellular metabolic hubs that play a key role energy metabolism and intracellular signalling. They are synthesised from two distinct genomes: nuclear DNA and mitochondrial DNA (mtDNA) and are assembled from ~5% of the cell proteome. Genetic variation of both mtDNA and nuclear DNA coding for mitochondrial proteins influences cell function and can contribute to human diseases. Severe mutations of both genomes cause monogenic mitochondrial disorders which affect~1/5000 and have a wide range of clinical presentations which overlap with common diseases and have distinct natural histories and recurrence risks. Reaching a precise genetic diagnosis has important implications for clinical management, with new treatments emerging for specific genetic groups. More subtle polymorphic variation of mtDNA interacts with the nuclear genome and the environment to modify the risk of developing several complex traits, and genetic variation of both nuclear and mtDNA can alter the risk of developing adverse drug reactions. These observations highlight the potential impact of incorporating analysis of genes coding for the mitochondrial proteome, but the field is in its infancy and ripe for further development.

## Introduction

Mitochondria are ubiquitous compartments present in all nucleated mammalian cells. They are the principal source of intracellular energy in the form of adenosine triphosphate (ATP) which is generated through oxidative phosphorylation (OXPHOS). Although ATP can also be generated anaerobically by glycolysis, aerobic OXPHOS produces far more ATP. This is essential for anabolic processes including protein synthesis and cell division, and also physiological functions including neuronal firing, cardiac and skeletal muscle contraction, hormone biosynthesis and secretion. In addition to their role in energy metabolism, mitochondria also act as metabolic hubs modulating and controlling multiple cellular mechanisms including calcium signalling, and as key partners in programmed cell death (apoptosis). As a consequence, mitochondria are central to human physiology and therefore, not surprisingly, have been implicated in the pathogenesis of many different human diseases (Vafai and Mootha, 2013; Wallace, 2018).

### Mitochondrial biogenesis

Mitochondria are currently estimated to incorporate over 1,100 proteins (Rath et al., 2021). The vast majority of these are encoded by the nuclear genome, but 13 are synthesised from small circles of double-stranded DNA present within the mitochondrial matrix: mitochondrial DNA (mtDNA, Figure 1). The nuclear gene products include structural sub-units involved in OXPHOS that physically interface with mtDNA-encoded components to deliver ATP synthesis.

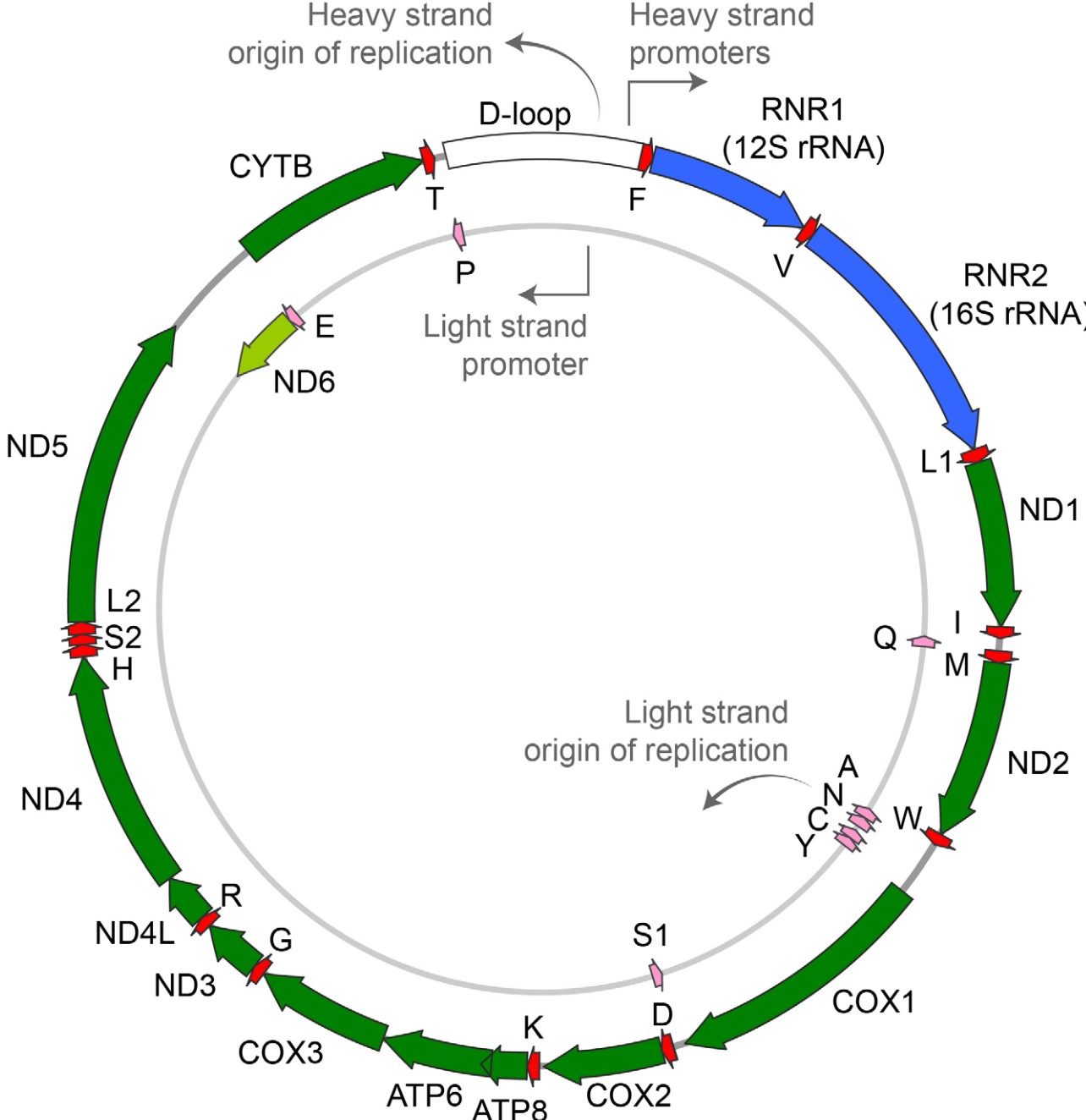

**Figure 1.** Human mitochondrial DNA (mtDNA). The 16,569 base pair human mtDNA includes an inner 'light' (L)-strand and an outer 'heavy' (H)-strand. mtDNA genes encoding structural sub-units of the mitochondrial respiratory chain include *ND1–ND6* and *ND4L* (complex I), *CYB* (complex III), *CO1–3* (complex IV), and *ATP6* and *ATP8* (complex V). The 22 tRNA and 2 rRNA genes are interspersed between the peptide-encoding genes and are essential for protein synthesis within mitochondria (amino acid letter codes). mtDNA replication is initiated by transcription within the non-coding mtDNA displacement (D) loop and proceeds from the origin of heavy-strand replication ($O_H$, also known as OriH) until the origin of light-strand replication ($O_L$) is exposed, allowing light-strand synthesis to proceed clockwise until the entire molecule is copied. Alternatively, symmetric strand-coupled replication might occur in certain circumstances. *Source*: Adapted from **Nature Reviews Genetics**, 16, 2015, 530–42, Springer Nature.

Additional nuclear proteins are involved in the maintenance of mtDNA, mtDNA transcription, intra-mitochondrial protein synthesis and the structural integrity of the organelle itself.

## Mitochondrial DNA

Each cell contains multiple copies of mtDNA (Figure 2) with the precise amount being tightly regulated and varying from cell type to cell type: at one extreme, sperm contain ~100 mtDNA molecules, whereas skeletal muscle fibres contain many tens of thousands of molecules. Human mtDNA is 16.5 Kb and codes for 13 structural sub-units of the OXPHOS system and 24 RNAs which play a key role in intramitochondrial protein synthesis (Stewart and Chinnery, 2015). Only a small 1.1 Kb segment of mtDNA is non-coding and involved in regulating replication and transcription of the mitochondrial genome, including a third strand of DNA called the displacement or 'D'-loop which incorporates the poorly understood 7S DNA. mtDNA has a different

**Wild-type mtDNA**  •  **Mutant mtDNA**

Biochemical threshold

**Figure 2.** mtDNA heteroplasmy and the threshold effect. Recent mtDNA mutations are usually heteroplasmic. Organs, cells and probably mitochondria can contain varying proportions of mutated and wild-type mtDNA. If a mutation is pathogenic, the cell can usually tolerate a high percentage level before the biochemical threshold is exceeded and a biochemical defect develops. The level of heteroplasmy can vary between individuals within the same family, and also change during life in some tissues and organs. *Source*: Adapted from Nature Reviews Genetics, 16, 2015, 530–42, Springer Nature.

amino-acid code to the nuclear genome and is exclusively maternally inherited in the general population. Although there have been recent reports of apparent paternal transmission of mtDNA (Luo et al., 2018), there are other likely explanations for these findings (Rius et al., 2019; Wei et al., 2020; Bai et al., 2021; Lutz-Bonengel et al., 2021) (discussed below).

### Heteroplasmy and the threshold effect

Mutations of mtDNA can affect a proportion of the molecules (heteroplasmy) usually reflecting in their recent origin. Most mtDNA mutations only affect OXPHOS when the proportion of mutated molecules exceeds a critical threshold value, typically greater than 50%. Several mutations only affect cellular biochemistry when greater than 75% of mtDNA molecules are affected (Figure 2; Stewart and Chinnery, 2015, 2020). Not only does the biochemical threshold vary from mutation to mutation, but also from cell type to cell type. The level of heteroplasmy can vary from cell to cell, organ to organ and from individual to individual. The reasons for this are only partially being unravelled. For some mtDNA mutations, the absolute amount of wild-type mtDNA appears to determine whether a cell expresses a biochemical defect (i.e., the mutations are haploinsufficient; Durham et al., 2007), but this is not the case for others, where a very high percentage level of the mutation is required to have an effect (Boulet et al., 1992). Very high thresholds probably reflect functional complementation by the residual wild-type molecules, which is only compromised when the proportion of mutant mtDNA is extremely high and wild-type levels correspondingly very low (Yoneda et al., 1994; Attardi et al., 1995). If all of the mtDNA is identical, this situation is termed 'homoplasmy'.

### Maternal inheritance and the genetic bottleneck

Profound differences in mtDNA heteroplasmy levels are often seen amongst siblings within the same maternal pedigree (Lightowlers et al., 1997). Similar observations in Holstein cows led to the mtDNA genetic bottleneck hypothesis (Ashley et al., 1989), whereby a reduction in the amount of mtDNA passed from mother to child leads to a statistical sampling effect, with different proportions of mutant and wild-type mtDNA being passed on to each offspring (Figure 3). Recent evidence across several species supports the existence of a physical reduction in the amount of mtDNA passed from one generation to the next (Cree et al., 2008; Wai et al., 2008; Otten et al., 2016; Floros et al., 2018). The key steps occur during early oocyte development in the developing female embryo, when the amount of mtDNA falls to ~200 molecules in each primordial germ cell (PGC) in mice (Cree et al., 2008; Wai et al., 2008), and ~500 copies in human PGCs (Floros et al., 2018). *In silico* modelling has shown that this reduction is sufficient to cause a sampling effect leading to very different levels of heteroplasmy in primary oocytes generated from PGCs (Cree et al., 2008), thus

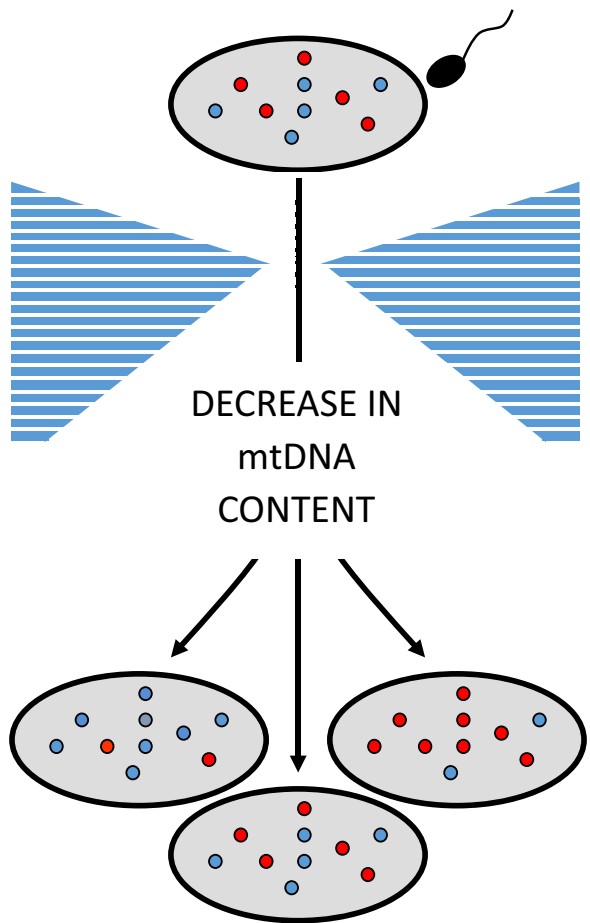

DECREASE IN mtDNA CONTENT

**Figure 3.** mtDNA genetic bottleneck. A reduction in the amount of mtDNA passed from one generation to the next leads to a statistical sampling effect and very different proportions of mutant and wild-type mtDNA in the offspring of the next generation. *Source*: Adapted from Nature Genetics 40, 2008, 249–54, Springer Nature.

determining the level of heteroplasmy passed on to the next generation. Selective forces acting for and against propagation of specific mtDNA mutations can modulate this further (Stewart et al., 2008; Freyer et al., 2012; Sharpley et al., 2012). Similar genetic bottlenecks also occur probably in somatic tissues contributing to different levels of heteroplasmy in different organs (Cao et al., 2007; Floros et al., 2018; Barrett et al., 2020; Tang et al., 2022).

### Precision diagnosis of rare mitochondrial disorders

Our first insight into the role of mitochondrial genetic variants in precision medicine arose through the investigation of patients with rare inherited primary mitochondrial diseases.

**Table 1.** Canonical mitochondrial disease clinical syndromes

|  | Primary features | Additional features |
| --- | --- | --- |
| Alpers–Huttenlocher syndrome | Childhood onset refractory seizures, developmental delay and liver failure | Hypotonia, ataxia, sensitivity to valproate |
| Chronic progressive external ophthalmoplegia (CPEO) | External ophthalmoplegia and bilateral ptosis | Proximal myopathy |
| Infantile myopathy and lactic acidosis (fatal and non-fatal forms) | Hypotonia in the first year of life<br>Feeding and respiratory difficulties | Cardiomyopathy and/or the Toni–Fanconi–Debre syndrome |
| Kearns–Sayre syndrome (KSS) | PEO onset before age 20 with pigmentary retinopathy plus one of the following: CSF protein greater than 1 g/l, cerebellar ataxia, heart block | Bilateral deafness<br>Myopathy<br>Dysphagia<br>Diabetes mellitus and hypoparathyroidism<br>Dementia |
| Leber hereditary optic neuropathy (LHON) | Subacute painless bilateral visual failure<br>Males:females approximately 4:1<br>Median age of onset 24 years | Dystonia |
| Leigh syndrome (LS) | Subacute relapsing encephalopathy with cerebellar and brain-stem signs presenting during infancy | Basal ganglia abnormalities |
| Mitochondrial neurogastrointestinal encephalomyopathy (MNGIE) | Chronic progressive external ophthalmoplegia, ptosis, gastrointestinal dysmotility (pseudo-obstruction), peripheral neuropathy, and myopathy | Diffuse leukoencephalopathy |
| Mitochondrial encephalomyopathy with lacticacidosis and stroke-like episodes (MELAS) | Stroke-like episodes before age 40 years<br>Seizures and/or dementia<br>Ragged-red fibres and/or lactic acidosis | Diabetes mellitus<br>Cardiomyopathy<br>Bilateral deafness<br>Pigmentary retinopathy<br>Cerebellar ataxia |
| Myoclonic epilepsy with ragged-red fibres (MERRF) | Myoclonus<br>Seizures<br>Cerebellar ataxia<br>Myopathy | Dementia, optic atrophy<br>Bilateral deafness<br>Peripheral neuropathy<br>Spasticity<br>Multiple lipomata |
| Neurogenic weakness with ataxia and retinitispigmentosa (NARP) | Late childhood or adult-onset peripheral neuropathy with associated ataxia and pigmentary retinopathy | Basal ganglia abnormalities<br>Abnormal electroretinogram<br>Axonal sensorimotor neuropathy |
| Pearson syndrome | Sideroblastic anaemia<br>Pancytopenia<br>Exocrine pancreatic failure | Renal tubular defects |

*Source*: Adapted from Pfeffer and Chinnery (2013).

### Primary mitochondrial disorders – definition and prevalence

Primary mitochondrial disorders are presumed genetic disorders leading to a primary defect of OXPHOS and ATP synthesis (Gorman et al., 2016). It is important to distinguish these from the many different mitochondrial biochemical abnormalities described as a secondary phenomenon in many common and rare disorders. Genetic epidemiology studies have shown that primary mitochondrial disorders affect ~1 in 4,300 of the general population (Gorman et al., 2015).

### Clinical heterogeneity

Given the central role of mitochondrial ATP production in many different organ systems, it is perhaps not surprising that mitochondrial disorders can affect multiple different organ systems. Typically, they involve tissues and organs with a high ATP dependence including the brain, hearing, visual and neuromuscular systems, the heart and endocrine glands including pancreatic islets (Gorman et al., 2016). The clinical overlap with common disorders, coupled with the fact that different individuals in the same family may have different organs and tissues involved, has made the clinical diagnosis of mitochondrial disorders, particularly challenging. As a rule of thumb, people who have a multisystem disorder with some neurological involvement and no obvious explanation should be considered to have a possible mitochondrial disorder. Several classical mitochondrial clinical syndromes have been described (Table 1), but many patients do not share all of the clinical features defined by canonical syndromic diagnoses. Intriguingly, some mitochondrial disorders are highly organ specific, causing sensorineural deafness, or visual impairment due to involvement of the retinal ganglion cell. However, many mitochondrial disorders involve multiple organ systems with many degrees of severity. Many children have multisystem mitochondrial disorders that do not fit neatly into one of the previously described clinical syndromes. Several mechanisms have been proposed to explain tissue selectivity and clinical heterogeneity, including differences in the burden of mtDNA mutations between tissues and organs (heteroplasmy, see above), and the existence of different tissue-specific isoforms of the nuclear-encoded OXPHOS sub-units – but our understanding is far from complete.

## Genetic basis of mitochondrial disorders

Inherited mitochondrial disorders are caused by mutations in either nuclear genes encoding for mitochondrial proteins or mtDNA. The first pathogenic mutations of mtDNA were identified three decades ago. Three different homoplasmic missense mutations in respiratory chain genes encoding for complex 1 sub-units have been identified in families with Leber's Hereditary Optic Neuropathy (LHON: m.11778A>G, m.3460A>G and m.14484T>C; Wallace et al., 1988; Howell et al., 1991). LHON is a maternally inherited disorder that causes sub-acute visual failure presenting in mid-adult life. It remains the most common mitochondrial disease worldwide, but is unusual, because the phenotype is highly tissue specific and the underlying mtDNA mutations have a striking incomplete penetrance: the ratio of affected males to females is approximately 4:1, with 40% of male mutation carriers and ~10% of females developing symptoms (Carelli et al., 2004). Early on it was recognised that a particular mitochondrial genetic background was associated with the disorder (Brown et al., 1995), and recent population studies have shown that common polymorphisms influence the clinical penetrance of the underlying pathogenic mtDNA mutation (Hudson et al., 2007). Remarkably, LHON mtDNA mutations are found in ~1:300 of the population (Elliott et al., 2008) and both environmental and additional genetic interactions likely play a key role in clinical expression, most notably cigarette smoking and heavy alcohol use (Kirkman et al., 2009). Thus, LHON has set the paradigm for gene–environment interactions in mitochondrial disorders and provides evidence of epistasis between different genetic variants within the mtDNA, and possibly the nuclear genome.

Around the same time, the first heteroplasmic pathogenic mutations were identified. Mutations in the transfer RNA gene for Leucine UUR were described in families with mitochondrial encephalomyopathy with lactic acidosis and stroke-like episodes (MELAS; Goto et al., 1990). Although subsequent work has shown that the full clinical syndrome is actually uncommon, the same mutation has been associated with maternally inherited diabetes, an eye movement disorder called chronic progressive external ophthalmoplegia (CPEO), and/or maternally inherited diabetes and deafness. The m.3243A>G MTTL1 mutation remains the most common heteroplasmic mtDNA mutation presenting in the clinic (Ng et al., 2021).

Finally, large-scale deletions of mtDNA were found to encompass key mitochondrial structural genes and transfer RNA genes causing CPEO either in isolation or in combination with other central neurological features, diabetes and cardiac induction block as a part of the Kearns–Sayre syndrome (Zeviani et al., 1988; Moraes et al., 1989). Unlike the previous two disorders, large-scale mtDNA deletions are only very rarely inherited and usually caused a sporadic disorder (Chinnery et al., 2004).

The last three decades have seen a huge increase in the number of different point mutations and deletions of mtDNA associated with primary mitochondrial disorders, some with overlapping phenotypes and some associated with distinct clinical syndromes which are either tissue specific or involve multiple different organ systems.

## Nuclear gene mutations causing mitochondrial diseases

The first likely nuclear genetic disorders causing mitochondrial diseases were identified in 1989. Families with autosomal dominant CPEO showing clear male-to-male transmission were described with multiple secondary deletions of mtDNA in skeletal muscle (Zeviani et al., 1989). This raised the concept of 'disorders of mtDNA maintenance', with the underlying nuclear genetic basis for disorders being identified several years later (Spelbrink et al., 2001; Van Goethem et al., 2001). These findings opened up the field to identify further nuclear genetic mitochondrial disorders, including defects affecting mtDNA maintenance, individual respiratory chain sub-units or their assembly, genes maintaining mitochondrial structural integrity or involved in fission and fusion of mitochondria, and genes involved in the intra-mitochondrial protein synthesis (Wallace, 2018; Gusic and Prokisch, 2021). Initial candidate gene and genetic linkage studies were superseded by next-generation sequencing approaches involving panels (Lieber et al., 2013), exome sequencing (Taylor et al., 2014), transcriptomics (Stenton and Prokisch, 2020a, 2020b) and latterly whole-genome sequencing (WGS) (Riley et al., 2020; Schon et al., 2021). The number of genes identified in families with biochemical evidence of mitochondrial disease continues to increase, with ~350 identified to date (Schon et al., 2020). This falls far short of the predicted number of different mitochondrial proteins (~1,100) indicating there are further disease genes yet to discover, although it is possible that some of these genes have such a catastrophic effect that we will never see them present in the clinic.

## Investigation of suspected mitochondrial diseases

The traditional approach for investigating mitochondrial diseases involves assimilating a portfolio of clinical evidence supplemented by the investigation of vulnerable organ systems, followed by a tissue biopsy leading to histochemical and/or biochemical investigation of the OXPHOS system (Taylor et al., 1998). This approach requires the involvement of specialist laboratories, is time consuming, expensive and can lead to protracted diagnostic odyssey (Grier et al., 2018). It is also not completely definitive, because many individuals with known mitochondrial disorders do not have detectable biochemical abnormalities using conventional diagnostic tests. The particular pattern of histochemical or biochemical defect has helped target molecular genetic analyses, which was particularly valuable when the capability for high-throughput sequencing was limited. This has changed dramatically over the last 5 years as explained below.

### The added complexity of mtDNA heteroplasmy

Tissue differences in mtDNA heteroplasmy can present a challenge for molecular diagnosis. Non-dividing (post-mitotic) cell tissues typically have higher levels of heteroplasmic mtDNA mutations than dividing cells. At its most extreme, for the most common heteroplasmic mutation (m.3243A>G) the level of heteroplasmy decreases in blood over time, probably because of selection against the pathogenic variant at the stem cell level (Pyle et al., 2007; Rajasimha et al., 2008). This means that the heteroplasmy level can fall below the detection threshold for some molecular tests. This can be circumvented by studying skeletal muscle or other tissue such as urinary epithelial cells (McDonnell et al., 2004). For some mitochondrial disorders, particularly those caused by single large-scale mtDNA deletions, the underlying gene defects are undetectable in blood. The important clinical point here is that absence of an mtDNA mutation from blood does not exclude a high level in clinically relevant tissues, and clinicians suspecting a mitochondrial disorder should not stop their investigations if the initial molecular genetic tests come back negative.

## Pitfall of pseudo-heteroplasmy

Segments of mtDNA have translocated from the organelle into the nucleus over evolutionary time (Dayama et al., 2014). The majority of these reside in non-coding space and are epigenetically silenced. These so-called 'nuclear mitochondrial sequences' (NUMTS) can introduce a confounder in a molecular diagnostic pathway, resembling true mtDNA heteroplasmy (Wei et al., 2020). This is the likely explanation for the recent description of apparent paternal transmission of mtDNA, which is more likely to be due to the transmission of NUMTS down the paternal line creating the impression of a heteroplasmic variant transmission (Rius et al., 2019; Wei et al., 2020; Bai et al., 2021; Lutz-Bonengel et al., 2021). Although in theory, this could lead to misdiagnosis of mtDNA diseases, in practice this is unlikely. In a recent survey of NUMTS in over 66,000 humans, there was no evidence that nuclear NUMTS sequences harboured variants resembling known pathogenic mtDNA mutations (Wei et al., 2022). Nonetheless, this is a potential confounder that should be considered particularly during the exploration of new genetic causes for mitochondrial diseases. There are both molecular and bioinformatic approaches to minimise NUMTS contamination of mtDNA sequencing.

## Genomic diagnosis

Standard diagnostic WGS approaches re-sequence an individual's genome 40- to 60-fold to optimise the chance of detecting pathogenic variants with confidence. Given the very high copy number of mtDNA in blood, this means that standard WGS also provides a complete mtDNA sequence at ~2,000-fold, enabling the detection of heteroplasmic variants down to the variant allele frequency (VAF, or heteroplasmy level) ~1% (Wei et al., 2019). The retrospective analysis of confirmed diagnoses of mitochondrial disease indicates that blood WGS is likely to detect over 90% of known pathogenic mutations of mtDNA and nuclear DNA, providing a strong argument that blood WGS should be the first-line diagnostic test (Raymond et al., 2018). For the reasons explained above, in the absence of a clear diagnosis from WGS, the standard diagnostic algorithm is appropriate with the added benefit of providing biochemical functional evidence to support the molecular basis of a novel genetic variant or novel gene defect.

## Current diagnostic yield and the implications of a negative test

Several studies have shown that the diagnostic yield of both exome and WGS depends on how carefully individual families are selected for investigation. The yield is greater when patients fulfil established diagnostic criteria (such as the Nijmegen criteria; Morava et al., 2006), particularly if this involves the biochemical confirmation of an underlying respiratory chain enzyme deficiency (Lieber et al., 2013; Taylor et al., 2014; Riley et al., 2020; Davis et al., 2022; Yepez et al., 2022). Perhaps the closest estimate of the true diagnostic yield in relatively unselected patients comes from the 100,000 Genomes Project in England, where patients with a suspect mitochondrial disorder were referred for WGS from both non-specialist and specialist centres (Schon et al., 2021). It is important to note that patients were only referred *after* undergoing conventional genetic investigations for a mitochondrial disorder. In this study, WGS delivered a molecular diagnosis in an additional 31% of families, with the highest yield in children, and particularly when parental DNA samples were available. However, over half of the new diagnoses were not classical mitochondrial diseases, emphasising the challenge of clinically defining a mitochondrial disorder up front. As mentioned above, WGS is capable of detecting >90% of currently known genetic causes of mitochondrial disease. Putting this information together, WGS has the potential to diagnose the vast majority of known mitochondrial diseases, approaching ~95% of referrals from secondary (hospital) care. However, it is important to note that patients with a positive genetic diagnosis do not fulfil clinical diagnostic criteria, and not all have a detectable biochemical abnormality, so it would be inappropriate to exclude patients from the opportunity to undergo exome or WGS in the absence of a classical clinical presentation.

There are several reasons why genetic testing can be negative in mitochondrial disorders: (i) for a mtDNA mutation, the level of heteroplasmy may be undetectable in the tested tissue (e.g., mtDNA deletions may not be detected in blood DNA, but reach high levels in skeletal muscle causing disease); (ii) the underlying causal nuclear gene may not be discovered yet; (iii) structural variants or repeat sequences may be missed using current diagnostic and bioinformatic pipelines; and (iv) detected variants may be difficult to interpret, perhaps lying in deep intronic regions. Thus, negative genetic testing does not exclude the diagnosis of a mitochondrial disorder. However, our capability to provide a WGS-based diagnosis for the mitochondrial disease has been greatly enhanced by the growing global data set of whole mtDNA and whole nuclear genome sequencing which facilitates interpretation (Wei et al., 2019; Laricchia et al., 2022), and this will undoubtedly improve in the coming years.

## Importance of a genomic diagnosis for mitochondrial diseases

Reaching a genomic diagnosis has important implications for families with mitochondrial disease.

- *Prognosis and surveillance.* The advent of molecular testing has allowed natural history studies of genetically defined groups at scale across the globe. This has provided an insight into prognosis and likely complications that can be effectively managed in individual patients. Typically this includes surveillance for cardiomyopathy, hearing loss and diabetes mellitus – all of which can be managed with standard approaches including transplantation and implantation.

- *Prevention.* The different molecular diagnoses have profoundly different implications for the family, including a sporadic disorder (large-scale mtDNA deletion), maternal inheritance (pathogenic mtDNA mutation), or dominant recessive and x-linked inheritance. It is important to emphasise that, from a clinical perspective, it may be impossible to distinguish these options at the bedside. However, a molecular diagnosis enables reliable genetic counselling in each context and pre-natal or pre-implantation diagnosis. For mtDNA mutations, there is the added opportunity of mitochondrial transfer to prevent transmission (Hyslop et al., 2016; Kang et al., 2016; Yamada et al., 2016; Chinnery, 2020), which has been successfully used to prevent Leigh syndrome (Hamzelou, 2016).

- *Mechanisms and treatments.* A precise genetic diagnosis enables targeted treatments such as co-enzyme Q10 and analogues in defects of ubiquinone biosynthesis (Yubero et al., 2018), or riboflavin supplementation in patients with *SLC52A2* mutations (Cali et al., 1993). Several gene therapy approaches are also in development and require a precise molecular diagnosis (Di Donfrancesco et al., 2022). Disease gene discovery for mitochondrial disorders has also identified many molecular pathways essential for mitochondrial biogenesis and maintenance. Many of these are attractive treatment targets that are currently being explored in pre-clinical studies, or in some instances through experimental medicine at early phase studies, often in partnership with industry (Nightingale et al., 2016).

## Genomic variants affecting mitochondrial function and common diseases

Primary genetic mitochondrial disorders share many clinical features with common human diseases, including Parkinson's disease (PD), dementia, diabetes and hypertension (Gorman et al., 2016). This raises the possibility that genetic variants with less severe or catastrophic effects could be present in the population as polymorphisms and affect the risk of developing common complex late-onset human disorders (Stewart and Chinnery, 2015). Animal models with specific primary mitochondrial disorders share some of these phenotypes including features of premature ageing (Kujoth et al., 2005). These combined observations opened up a field of enquiry that is yielding results likely to be important for precision medicine.

### *Role of common nuclear genomic variants in common diseases*

Building on the observation of specific phenotypes seen in rare primary mitochondrial disease patients, candidate nuclear gene studies have been performed to determine whether or not polymorphic variants in the same gene might be genetic risk factors for common disorders. There are several examples in the literature, perhaps most notably in *POLG*, the gene coding for the only mtDNA polymerase Y. Common polymorphic variants and heterozygous pathogenic mutations have been reported in association with late-onset idiopathic PD (Luoma et al., 2007), although not all studies have reproduced these findings (Hudson et al., 2009). Likewise, the 16184–16193 poly-C length variant in the mtDNA non-coding region has been associated with type 2 diabetes in specific populations (Poulton et al., 2002), but not others (Chinnery et al., 2005). The majority of candidate gene studies have been underpowered for the effect size being reported, questioning the strength of the original findings, which often did not incorporate well-known nuclear genetic risk loci in the same analyses. Although mitochondrial genes have fallen within high-risk loci mapped for common disorders in hypothesis-free genome-wide association studies (as for *ATP5G1* in type 2 diabetes) (Xue et al., 2018), a recent meta-analysis found no evidence of enrichment for mitochondrial genes in genome-wide association data for 24 age-related human traits (Gupta et al., 2021). There are therefore no convincing validated examples of specific nuclear alleles associated with common diseases within known mitochondrial genes. This is, perhaps, surprising given the previous arguments made in this review. However, the functional redundancy of essential metabolic pathways and the semi-autonomous nature of mitochondria are potential explanations (Gupta et al., 2021).

### Mitochondrial DNA

Ultimately *de novo* mtDNA mutations affect a single molecule and begin in a heteroplasmic state. Over time, the level of heteroplasmy can increase or decrease, with a rate of segregation accelerated by the germ line genetic bottleneck (see above). Ultimately a genetic variant may become fixed in the maternal line and thus be homoplasmic (Stewart and Chinnery, 2015). Both heteroplasmic and homoplasmic variants of mtDNA have been associated with disease (Stewart and Chinnery, 2020).

### *Population genetics of mtDNA*

The uniparental inheritance of mtDNA has been exploited by anthropologists to study human population migrations. As modern humans emerged from Africa ~150,000 years ago and spread around the globe, the different regional populations acquired different mtDNA variants defining so-called haplogroups (Wallace, 1994). The frequency of specific haplogroups can vary over relatively short geographical distances, likely due to genetic drift and population bottleneck effects that particularly affect less common genetic variants (Yonova-Doing et al., 2021). However, an analysis of over 30,000 human mtDNA sequences sampled from across the globe shows that the haplogroup defining mitochondrial DNA Single Nucleotide Variants (mtSNVs) are not randomly distributed across the genome (Wei et al., 2017). These findings implicate selected forces acting for and against genetic variants at particular sites based on their functional consequences. Biochemical studies using cybrid cell lines, which control for the underlying nuclear genetic background, add weight to the idea that common mtDNA polymorphisms can have functional effects on mitochondrial function which extend beyond OXPHOS (Gomez-Duran et al., 2010, 2012). Population genetic studies have cast further light on this.

### *Evidence haplogroups an effect in primary mtDNA disorders*

The strongest evidence that mtDNA haplogroups influence human disease comes from the investigations of LHON, which in Europeans is caused by one of three homoplasmic mtDNA mutations: m.11778G>A, m.14484T>C and m.3460A>G (see above) (Wallace et al., 1988; Howell et al., 1991; Johns and Berman, 1991). Shortly after their first definition, it was recognised that these mutations preferentially associate with mtDNA haplogroup J which is found in ~11% of Europeans (Torroni et al., 1997). Detailed mtDNA genotyping showed that the association was not due to founder effects, with the most marked association for m.14484T>C (Torroni et al., 1997). Recent population studies have shown that the mtDNA haplogroup likely influences the clinical penetrance of the underlying pathogenic mtDNA variant (Hudson et al., 2007; Yonova-Doing et al., 2021). Extensive biochemical analyses support evidence of biochemical interaction between the different genetic variants (Carelli et al., 1999; Gomez-Duran et al., 2012), adding weight to the concept that similar haplogroup-specific genetic variation could contribute to common human diseases where mitochondrial dysfunction is important.

### *Preliminary evidence that mtDNA haplogroups affect common disease risk*

Many candidate gene studies have been performed looking for evidence of association between mtDNA haplogroups and common disorders including asthenozoospermia (Ruiz-Pesini et al., 2000), type 2 diabetes and its complications (Kofler et al., 2009; Achilli et al., 2011; Estopinal et al., 2014; Bregman et al., 2017), coronary artery disease (Kofler et al., 2009; Mueller et al., 2011), hypertension (Zhu et al., 2009), multiple sclerosis (Ban et al., 2008), and neurodegenerative disorders including PD (Ghezzi et al., 2005) and Alzheimer's disease (Santoro et al., 2010). The majority of these studies were relatively small, with several hundred participants, and few involved direct replication (Samuels et al., 2006). Large-scale replication studies have been performed in a number of instances (Hudson et al., 2012, 2013a,2013b; Kraja et al., 2019), most notably in idiopathic PD where the consistent effect on disease risk has been identified, with haplogroup UK reducing risk and haplogroup J increasing risk. In addition, mtDNA variants have been convincingly associated with metabolic phenotypes in several populations, raising the possibility that mtDNA variation could be combined with known nuclear risk loci in, for example, polygenic risk scores.

### Population association studies of mtDNA variants

In a recent study, in 1928 people in the Japanese Biobank looked for associations between 2,023 mtDNA variants derived from whole-genome-sequence data to impute mtDNA variants in 147,437 Japanese individuals with 99 common traits (Yamamoto et al., 2020), and observed pleiotropy of mtDNA genetic risk on the five late-onset human complex traits including creatine kinase levels. A larger study of 473 mtDNA variants in 358,916 individuals in UK Biobank discovered 260 novel mtDNA-phenotype associations, including novel variants enriched in type 2 diabetes and multiple sclerosis (Yonova-Doing et al., 2021). There were also many mtDNA associations with common physiological parameters including biochemical markers of liver and renal function, and routine blood cell measures including the red blood cell count, mean corpuscular haemoglobin, mean red blood cell volume (including the m.11778A>G LHON mutation, see above), platelet count and platelet volume. mtDNA variants were also associated with height and longevity, including the common m.1555A>G variant discussed below. Not all previously replicated disease associations were confirmed in this study (e.g., PD), because even at this scale, the number of affected individuals with a particular disease category was small compared to the targeted disease association studies performed previously. Overall, these findings support the concept that common polymorphic variants of mtDNA influence human characteristics and disease phenotypes in combination with other genetic and environmental aetiological factors, mirroring phenotypes seen in patients with severe primary mitochondrial disorders who harbour genetic defects that have more profound biochemical effects (Stewart and Chinnery, 2015). The underlying mechanisms are starting to be unravelled, and likely extend beyond OXPHOS. Recent evidence implicates cell proteostasis mediated through the circulating metabolite N-formylmethionine which plays a central role in mitochondrial protein synthesis (Cai et al., 2021).

### Heteroplasmic variants

High-depth mtDNA sequencing that accompanies WGS has enabled population-level studies of mtDNA heteroplasmy and its transmission (Li et al., 2016). For example, there is recent evidence that heteroplasmic mtDNA variants contribute to cardiovascular disease risk through an effect on blood pressure (Calabrese et al., 2022). Some of these variants may emerge during life and possibly be a consequence of the disorder or its treatment, but some of the variants may have been maternally inherited, raising the possibility that heteroplasmic mtDNA variants contribute to the 'missing heritability' of some common diseases (Keogh and Chinnery, 2013). High levels of heteroplasmic mtDNA mutations have been detected in affected brain regions in PD (Bender et al., 2006), Alzheimer's disease (Corral-Debrinski et al., 1994) and frontotemporal dementia (Nie et al., 2022). Some of the mutations affect several brain regions, making it increasingly likely that they are maternally inherited, potentially contributing to the known increased risk of developing PD in maternal relatives. These findings are supported by mouse models where mtDNA heteroplasmy can contribute to premature ageing (Trifunovic et al., 2004; Kujoth et al., 2005).

### Mitochondrial genes in pharmacogenetics

There are a limited number of well-established associations between genetic variants in mitochondrial genes and drug-induced side effects. People with mitochondrial disease due to complex heterozygous mutations in the gene encoding the mtDNA polymerase Y (POLG) are susceptible to sodium valproate-induced liver toxicity, as part of the Alpers–Huttenlocher syndrome (Cohen et al., 1993). For this reason, sodium valproate should be avoided in patients with this particular genetic disorder. This has led many clinicians to avoid prescribing sodium valproate for all mitochondrial disorders, although clear evidence is lacking beyond POLG-related disease. Although we identified an association between common polymorphisms in POLG and unexplained drug-induced liver injury caused by sodium valproate (Stewart et al., 2010), these findings have not been replicated and are therefore tentative.

The m.1555A>G mtDNA mutation is carried by ~1:300 of the population (Elliott et al., 2008) and has been associated with aminoglycoside-induced hearing loss (Prezant et al., 1993; O'Sullivan et al., 2015). This has led many to screen for this mutation before beginning aminoglycosides including streptomycin and gentamycin, particularly in the intensive care setting. However, population studies do not provide strong evidence of a link between this variant and sporadic hearing loss (Kullar et al., 2016), suggesting that this may be a particular vulnerability seen in families with maternally inherited deafness, and not in the general population at large. Functional studies suggest that a deeper understanding of the mtDNA sequence will be informative in avoiding the complications of using other antibiotics (Gomez-Duran et al., 2011; Pacheu-Grau et al., 2013), but clinical data are lacking at present.

### Conclusions

Genomics has underpinned a diagnostic revolution enabling the precision diagnosis of rare inherited mitochondrial disorders. This underpins clinical management and is driving forward treatments, including at the molecular level. In this context, precise molecular diagnosis has a profound effect on the management of recurrence risks and disease surveillance, and thus the management of complications.

Although in its infancy, it is clear that genetic variation of mtDNA also plays an important role in determining many human physiological parameters and is a risk factor for common human diseases. Evidence linking nuclear-mitochondrial genes and common diseases are, however, limited. Precisely, how these findings feed into precision medicine strategies for common diseases have yet to emerge. Our understanding of mitochondrial pharmacogenomics is also likely to expand in coming years, given the recent evidence that canonical cell signalling pathways can be altered by genetic variants of both mtDNA, highlighting the central role of the mitochondrion in cell function. Although speculative at present, it is likely that understanding the genetic code responsible for the biogenesis of our mitochondria will influence how we use medicines in the future – to optimise treatment effects and minimise toxicity.

**Open peer review.** To view the open peer review materials for this article, please visit http://doi.org/10.1017/pcm.2022.8.

**Acknowledgements.** The views expressed are those of the author(s) and not necessarily those of the NIHR or the Department of Health and Social Care.

**Financial support.** P.F.C. is a Wellcome Trust Principal Research Fellow (212219/Z/18/Z), and a UK NIHR Senior Investigator, who receives support from the Medical Research Council Mitochondrial Biology Unit (MC_UU_00028/7), the Medical Research Council (MRC) International Centre for Genomic Medicine in Neuromuscular Disease (MR/S005021/1), the Leverhulme Trust (RPG-2018-408), an MRC research grant (MR/S035699/1) and an

Alzheimer's Society Project Grant (AS-PG-18b-022). This research was supported by the NIHR Cambridge Biomedical Research Centre (BRC-1215-20014).

**Competing interest.** The author declares no competing interests.

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
