## [Reviewer Report]

*Comments to Author*: The manuscript, authored by one of the leading scientists in the mitochondrial genetics field, elegantly and comprehensively covers basic mitochondrial biology, the genetics of mitochondrial DNA and mitochondrial protein-coding nuclear DNA, and their contribution to both primary mitochondrial and common complex disorders.

Such effort is welcome as the involvement of mitochondrial DNA variants in common complex diseases, not only in primary mitochondrial disorders, have been receiving increasing attentions for the last decades.

The topic is timely and clinically relevant. Indeed, with the increasing incidence of age-related complex diseases, where mitochondria play an increasingly recognised role, precision mitochondrial medicine is one pivotal field that can ultimately have a strong impact on our ageing society.

As such, the content is of great interest not only for specialists in "mitochondrial" medicine but also for general readers of the journal. This reviewer expects the manuscript to be heavily cited by many clinicians and scientists when published.

The structure of the manuscript is logical and easy to follow. Particularly, in the second section (2. Precision diagnosis of rare mitochondrial disorders), the hurdles of diagnosing primary mitochondrial diseases due to the variable clinical presentation as well as the complexity of mitochondrial gene variants (e.g., tissue specificity and the nature of mtDNA variants) are very well organised and nicely summarised. If a graphical summary of this part is available, it will make this manuscript more distinct as a precision mitochondrial medicine resource.

One interesting section of the manuscript that can be expanded is that dealing with gene-environmental interaction. The author primarily discussed this aspect in LHON. If there are any known interactions between environmental factors and non-pathogenic mtDNA polymorphisms in common complex diseases in humans and/or experimental models, it will support the concept.

Additionally, there are minor points that can be revised:

- The mouse models described in reference 60 do not exhibit primary mitochondria disorders, while these heteroplastic mice showed metabolic disease phenotypes. Therefore, this reference my not be adequate to the statement.

- Figure 2 looks identical to a figure published in PMID: 26281784 (presented as Figure 1 in the paper). Although this was published by the author of this manuscript, it may be a copyright issue?

- Figure 3 legend is to be checked for type (the beginning of the legend). Is "a statistical sampling affect" to be replaced as "a statistical sampling effect"? If so, the respective term in the main text (Page 3, line 4) should also be changed.

- Abbreviation VAF should be explained for general readers?

Overall, this reviewer strongly recommends this manuscript for publication.

---

## [Reviewer Report]

*Comments to Author*: Doctor Chinnery have made a comprehensive review regarding the relevance of the mitochondrial precision medicine mainly from the perspective of mtDNA but with significant references to nuclear encoded genes.

I enjoyed a lot reading it, but I have few suggestions that may improve the scope and significance:

1.- There is no mention to one of the earlier association of mtDNA variants to human disease regarding Astenozoospermia

2.- The updated catalog of mitochondria proteins is being proposed to rise to 1,300 proteins rather than 1,100

3.- It would be very interesting to comment why most if not all mtDNA mutations require to be accumulated well over 50% in heteroplasmy to cause deficiency. Does it mean that no dominant mutation in mtDNA has been described to date? Is there any reason for that?

4.- Beside the bottle-neck at the oocyte formation there are evidences of mtDNA variant selection during early embryo development. This need to be commented since it is relevant for diagnosis, treatment and prognosis.

5- In pag 11, last line the quoting seems to be incorrect since the 97 reference is not related with what is described.

6.- It would be of interest if the authors include a comment on the fact that the protein content of mitochondria of different cell types is different and on the existence of nuclear encoded isoforms for OXPHOS structural subunits and its potential impact on the modulation of mtDNA or nDNA mutations.

---

## [Editor Report]

*Comments to Author*: Dear authors, 

Comprehensive review on mitochondrial diseases was carried out in this article. Using simple and clear language, the text became very didactic even discussing complex information.

Enthusiastic mentions about molecular diagnosis of mitochondrial diseases were appropriately observed across the text. However, we would like to ask the authors about two points on precision molecular diagnosis. 

1. First, what is the genetic testing diagnosis yield in patients with previous clinical diagnosis of mitochondriopathy? Is there difference in genetic testing diagnosis yield between it versus in non selected (only clinical suspicious, no diagnosis clearly established) cohorts of patients with mitochondrial disease? What are the reasons for negative genetic results in patients with established clinical diagnosis of mitochondrial disease? It should be clear in the text, the genetic testing yield, and that a negative exome + mtDNA sequencing results do not rule out a clinical suspicious of mitochondrial disease.

2. Even though prenatal and pre-implantation diagnosis have been only cited in the Prevention topic, 2.5.4 section, it is a relevant information for clinical management and genetic counseling of these patients. No reference was cited here, please report it. Authors should add a sentence with the main disorders (or clinical scenarios) where these diagnostic approaches have been applied. 

Further, in the 3.2.3 section, the title describing "homoplasmic mtDNA variants..." does not clearly reflect the content of this section. There is no description in this text about homoplasmic status. The title or the text should be modified to make sense.